# In Silico Evaluation of HN-N07 Small Molecule as an Inhibitor of Angiogenesis and Lymphangiogenesis Oncogenic Signatures in Non-Small Cell Lung Cancer

**DOI:** 10.3390/biomedicines11072011

**Published:** 2023-07-17

**Authors:** Lung-Ching Chen, Ntlotlang Mokgautsi, Yu-Cheng Kuo, Alexander T. H. Wu, Hsu-Shan Huang

**Affiliations:** 1Division of Cardiology, Department of Internal Medicine, Shin Kong Wu Ho-Su Memorial Hospital, Taipei 11101, Taiwan; marcus1831@gmail.com; 2School of Medicine, Fu Jen Catholic University, New Taipei 24205, Taiwan; 3PhD Program for Cancer Molecular Biology and Drug Discovery, College of Medical Science and Technology, Taipei Medical University and Academia Sinica, Taipei 11031, Taiwan; d621108006@tmu.edu.tw; 4Graduate Institute for Cancer Biology & Drug Discovery, College of Medical Science and Technology, Taipei Medical University, Taipei 11031, Taiwan; 5Department of Pharmacology, School of Medicine, College of Medicine, Taipei Medical University, Taipei 11031, Taiwan; yuchengkuo@tmu.edu.tw; 6School of Post-Baccalaureate Chinese Medicine, College of Chinese Medicine, China Medical University, Taichung 40402, Taiwan; 7The PhD Program of Translational Medicine, College of Medical Science and Technology, Taipei Medical University, Taipei 11031, Taiwan; 8Clinical Research Center, Taipei Medical University Hospital, Taipei Medical University, Taipei 11031, Taiwan; 9TMU Research Center of Cancer Translational Medicine, Taipei Medical University, Taipei 11031, Taiwan; 10Graduate Institute of Medical Sciences, National Defense Medical Center, Taipei 11490, Taiwan; 11School of Pharmacy, National Defense Medical Center, Taipei 11490, Taiwan; 12PhD Program in Drug Discovery and Development Industry, College of Pharmacy, Taipei Medical University, Taipei 11031, Taiwan

**Keywords:** non-small cell lung cancer (NSCLC), bevacizumab, small molecule, hypoxia, vascular endothelial growth factor (VEGF), angiogenesis

## Abstract

Tumor angiogenesis and lymphangiogenesis pathways have been identified as important therapeutic targets in non-small cell lung cancer (NSCLC). Bevacizumab, which is a monoclonal antibody, was the initial inhibitor of angiogenesis and lymphangiogenesis that received approval for use in the treatment of advanced non-small cell lung cancer (NSCLC) in combination with chemotherapy. Despite its usage, patients may still develop resistance to the treatment, which can be attributed to various histological subtypes and the initiation of treatment at advanced stages of cancer. Due to their better specificity, selectivity, and safety compared to chemotherapy, small molecules have been approved for treating advanced NSCLC. Based on the development of multiple small-molecule antiangiogenic drugs either in house and abroad or in other laboratories to treat NSCLC, we used a quinoline-derived small molecule—HN-N07—as a potential target drug for NSCLC. Accordingly, we used computational simulation tools and evaluated the drug-likeness properties of HN-N07. Moreover, we identified target genes, resulting in the discovery of the target *BIRC5/HIF1A/FLT4* pro-angiogenic genes. Furthermore, we used in silico molecular docking analysis to determine whether HN-N07 could potentially inhibit *BIRC5/HIF1A/FLT4*. Interestingly, the results of docking HN-N07 with the *BIRC5, FLT4*, and *HIF1A* oncogenes revealed unique binding affinities, which were significantly higher than those of standard inhibitors. In summary, these results indicate that HN-N07 shows promise as a potential inhibitor of oncogenic signaling pathways in NSCLC. Ongoing studies that involve in vitro experiments and in vivo investigations using tumor-bearing mice are in progress, aiming to evaluate the therapeutic effectiveness of the HN-N07 small molecule.

## 1. Introduction

Globally, non-small cell lung cancer (NSCLC) is responsible for a significant number of cancer-related deaths and is the second most prevalent cause of cancer-related morbidity. It accounts for approximately 85% of newly diagnosed cases each year [1] and has an overall 5-year survival rate of below 15% [2]. Current treatment modalities include radiotherapy and chemotherapy, such as cetuximab and bevacizumab [3,4,5,6]. More than half of all NSCLC patients are diagnosed with local or distant metastatic disease or recurrence post-treatment, leading to drug resistance and poor clinical outcomes [7,8]. Therefore, there is an urgent need to elucidate the molecular mechanisms of NSCLC, as well as identify novel biomarkers, to ensure that novel drug targets can be developed [9,10]. Baculoviral IAP repeat containing 5 (BIRC5), which is also referred to as survivin, has been extensively studied in different cancer types, including in NSCLC [11,12,13]. *BIRC5* overexpression is associated with mitosis, proliferation, migration, and immune infiltration in different cancers; however, its clinical impacts and associations with the tumor microenvironment (TME) are still not well understood [14,15,16,17,18].

Since *BIRC5* is reported to be an immune-associated gene that promotes tumor progression, many studies related to survivin in NSCLC focus on the sensitization to chemotherapy, radiotherapy, and targeted therapy, with little success achieved due the heterogeneity of this disease [19,20]. Numerous studies have also shown that *BIRC5* overexpression is associated with angiogenesis and inhibition of cell apoptosis in lung cancer, thus making it a potential anti-NSCLC therapeutic biomarker [21,22,23,24]. As an angiogenic factor, BIRC5 sustains prolonged vascular endothelial cell integrity [25], and, hence, influences treatment responses in vascular diseases [26]. The growth of new vascular networks is crucial, since cancer progression depends on it for a sufficient supply of oxygen and nutrients [27]. In addition, tumor cells feed via newly formed blood vessels sourced from vascular networks, ultimately producing vascular endothelial growth factor (*VEGF*) and secreting it to nearby tissues [28,29,30]. There are five identified *VEGFs,* including *VEGF* (B,F, C, and D) and placental growth factor. These factors activate their specific receptors, including VEGF receptor-1, receptor-2, and receptor-3, as well as co-receptors, and play distinct roles in vascular development [27,31]. Previous studies showed that tumor progression and metastasis are mainly driven by important biological processes, such as angiogenesis and lymphangiogenesis [32,33].

Specifically, vascular endothelial growth factor C (*VEGF-C*) and its receptor *VEGFR*-*3,* which is also known as Fms-like tyrosine kinase 4 (*FLT4*), are expressed in tumor cells and associated with growth of blood vessels within tumors, as well as the progression of cancer cells to other parts of the body and the overall prognosis of the disease. *FLT4* is mainly expressed in the lymphatic endothelium and plays a crucial role in lymphangiogenesis and metastasis in malignant tumors [34,35,36]. Moreover, *FLT4* expression is developmentally regulated and mainly restricted to lymphatic endothelial tissues [37,38]. Therefore, it can serve as a marker of lymphatic endothelial cells [39,40,41,42,43]. *FLT4* expression was identified in various cancers; however, its involvement in NSCLC remains elusive, hence it is necessary to investigate its molecular mechanism in this disease [44,45,46]. Therefore, exploring the inhibition of the *VEGFR/FLT4* pathway may offer a promising strategy for preventing tumor lymphangiogenesis and metastasis in NSCLC [47,48]. Furthermore, research has indicated that hypoxia-inducible factor (*HIF)-1α,* which is a transcription factor, may have a significant impact on tumor growth and metastasis through its regulation of angiogenesis and lymphangiogenesis, which enables cellular survival in hypoxic conditions [49]. Additionally, hypoxia was found to activate *HIF-1α* in the NSCLC TME, leading to *VEGF* expression [50]. This process results in distant metastases and poor prognoses of NSCLC [51]. These findings suggest that crosstalk occurs among *BIRC5/HIF1α/FLT4,* thus driving lymphangiogenesis and metastasis in NSCLC.

Small-molecule targeted therapies have attracted increased interest in recent years, and they have become mainstream cancer treatments due to meeting conventional treatment modalities, including being safe, relatively cheap, and selective and specific with low minimal side effects. Moreover, these targeted drugs can effectively block molecular transduction pathways, activate immune responses, and induce apoptosis [52,53,54]. In the present study, we evaluate the anticancer activities of a quinoline-derived small-molecule compound—HN-N07—that was synthesized in our laboratory [55,56,57,58]. The compound was first sent to the national cancer institute (NCI) to be screened for its potential anticancer activities. Accordingly, we discovered that HN-N07 exhibited antiproliferative and cytotoxic effects on a panel of NSCLC cell lines at an initial single dose of 10 Μm, as well as in a dose-dependent manner [59,60,61]. Moreover, we utilized bioinformatics tools and discovered that BIRC5, HIF1α, and FLT4 oncogenes are target genes of HN-N07, among other genes. Furthermore, the results of molecular docking revealed that the compound displayed good binding energies, with a shorter binding distance needed when in complex with *BIRC5/HIF1α/FLT4* genes, suggesting that it could be a potential inhibitor of this signaling pathway in NSCLC.

## 2. Methods and Materials

### 2.1. Differential Expression of BIRC5/HIF1A/FLT4 in Normal, Tumor, and Metastatic Samples and a Correlation Analysis

Expression profiles of *BIRC5/HIF1A/FLT4* signaling were analyzed in normal tissues and compared to tumor tissues, as well as metastatic tissues, using TNMplot (https://tnmplot.com/analysis/, 28 November 2022). Accordingly, the database contained a total of 56,938 samples, which consisted of both RNA-seq and gene array samples. After undergoing pre-processing, there were 34,350 distinct gene array samples available, which covered 40 different tissue types. Among these samples, there were 3781 normal samples, 30,276 tumorous samples, and 462 metastatic samples. Data analyzed from selected oncogenes were based on rapid RNA-Sequencing (RNA-Seq) profiling, and the Kruskal–Wallis test was used to compare results [62]. Furthermore, we explored GEPIA2 (http://gepia2.cancer-pku.cn/, 28 November 2022) to analyze correlations between *BIRC5/HIF1A/FLT4* oncogenic signatures in NSCLC. A statistically significant association was determined based on positive Pearson’s correlation coefficients and a *p*-value of less than 0.05.

### 2.2. Validation of Pathological Stages of BIRC5/HIF1A/FLT4 Oncogenes in NSCLC and Their Prognostic Relevance

Pathological stage plots of *BIRC5/HIF1A/FLT4* in NSCLC were analyzed using GEPIA2 online software, with the major stage option used for plotting, and all data were selected from lung adenocarcinoma (LUAD) datasets, with log_2_ and transcripts per million +1 (TPM+1) used for the log-scale. Furthermore, we explored The Cancer Genome Atlas (TCGA) database sourced from the UALCAN online platform (http://ualcan.path.uab.edu/15, December 2022) to analyze *BIRC5/HIF1A/FLT4* expression in LUAD based on histological subtypes from TCGA samples, and these included LUAD-not otherwise specified (NOS), lung clear cell adenocarcinoma (clear cell), predominant adenocarcinoma with a solid pattern in the lungs, mucinous lung bronchoalveolar carcinoma (LBC-mucinous), lung papillary adenocarcinoma (papillary), lung micro-papillary adenocarcinoma (micro-papillary), lung adenocarcinoma with mixed subtypes (mixed), non-mucinous lung bronchoalveolar carcinoma (LBC), lung acinar adenocarcinoma (acinar), mucinous (colloid), and lung signet ring adenocarcinoma (signet ring). In addition, we determined overall survival (OS) in relation to expression of *BIRC5/HIF1A/FLT4* oncogenes using GEPIA2 software, with a significance level of *p* < 0.05 considered to be statistically significant.

### 2.3. Protein–Protein Interaction (PPI) Network Construction and Gene Enrichment Analysis (GEA)

Protein interactions were constructed using STRING (https://string-db.org/ 21 December 2022) [63]. For further exploration, we used the enriched PPI clustering networks from the STRING results to perform a GEA, using DAVID (https://david.ncifcrf.gov/.jsp, 22 December 2022), Funrich, and Network Analyst software to construct the graphs. The minimum level of significance was set at *p* < 0.05.

### 2.4. Analysis of BIRC5/HIF1A/FLT4 Genetic Mutations in LUAD Solid Tumors

Associations between genetic mutations of the *BIRC5/HIF1A/FLT4* oncogenes and their altered expression in LUAD were analyzed using the online mutation target (muTarget) bioinformatics tool (https://www.mutarget.com/, 4 January 2023) [64]. Herein, statistical significance was set at *p* < 0.05, helping us to determine the differences in expression levels between the mutant and wild-type (WT) groups. Moreover, we utilized the oncoprint webtool, which is embedded in cBioPortal software (https://www.cbioportal.org, 4 January 2023), to further analyze genetic alterations and copy number variations (CNVs) in the *BIRC5/HIF1A/FLT4* genes in LUAD, which were based on mutation spectra and alteration frequencies.

### 2.5. Analysis of scRNA-Seq Datasets Was Performed to Profile the Tumor Microenvironment (TME) in Primary and Metastatic Sites of Non-Small Cell Lung Cancer (NSCLC)

scRNA-Seq is widely used to study communication between cells and their TME, and Single-cell RNA sequencing (scRNA-Seq) offered a comprehensive approach that we used to gain deeper insights into the diverse populations within the tumor microenvironment (TME). It enabled the identification of novel cell types and the exploration of previously unknown associations within the TME [65,66].

Herein, we explored the Tumor Immune Single-cell Hub (TISCH), which is a powerful web-tool designed to comprehensively dissect the intricate components of the tumor microenvironment (TME) at a single-cell resolution. It provided a comprehensive and user-friendly platform through which to analyze and explore the complex characteristics of the TME at the cellular level [67].

### 2.6. Correlation Analysis of Immune Cell Infiltration and BIRC5/HIF1A/FLT4 Expression

We utilized the Tumor Immune Estimation Resource (TIMER 2.0, http://timer.cistrome.org/1, 4 January 2023) to examine the associations between the expression levels of *BIRC5*, *HIF1A*, and *FLT4* and the extent of tumor infiltration. This tool allowed us to analyze and assess the correlations between these examples of gene expression and the immune cell infiltration within the tumor microenvironment [68]. Herein, we mainly analyzed correlations between *BIRC5/HIF1A/FLT4* and infiltration of cancer-associated fibroblasts (CAFs) in LUAD. In addition, we analyzed mutations of *BIRC5/HIF1A/FLT4* in CAFs using the mutation module from TCGA clinical outcomes in the TIMER 2.0 algorithm. For further analysis, we determined distributions of *BIRC5/HIF1A/FLT4* expression in LUAD across different molecular subtypes, including wound healing, interferon (*IFN*)*-γ* dominant, inflammation, lymphocyte, and transforming growth factor (*TGF*)*-β* subtypes using TISIDB (http://cis.hku.hk/TISIDB, 20 February 2023) [69].

### 2.7. In Silico Flow Cytometric Analysis Using NSCLC Single RNA-Seq Bulk Tumors

To profile the infiltration of bulk RNA-Seq expression tumors, we utilized CIBERSORTx, which is a versatile online analytical tool (https://cibersortx.stanford.edu/, 25 February 2023) that allowed us to extract a signature from single-cell RNA-Seq data and quantify cell fractions sourced from differential bulk tumor gene expression [70].

### 2.8. Computational Evaluation of the Drug Likeness, and Properties of HN-N07

To determine the physicochemical properties, drug likeness, pharmacokinetics (PKs), physicochemical properties, and medicinal chemistry of HN-N07, we used SwissADME software, and to create target predictions, we used DTP-COMPARE algorithms [71]. These techniques were used to determine activity patterns of HN-N07 relative to its correlation with the National Cancer Institute (NCI) synthetic compounds and standards agents. For further analysis, we used Swisstarget software (http://www.swisstargetprediction.ch, 24 March 2023), which applies similar prediction principles to predict drug targets based on a “probability” target score [72].

### 2.9. Receptor–Ligand Binding Interaction Predictions through an In Silico Molecular Docking Analysis

A docking analysis was conducted by examining interactions of three-dimensional (3D) structures of receptors and ligands at the lowest possible binding energy [73]. Accordingly, to predict possible interactions between HN-N07 and its targets, including *BIRC5/HIF1A/FLT4*, as previously predicted from DTP-COMPARE and Swisstarget software, we performed in silico molecular docking analysis of HN-N07 using the *BIRC5,* FLT4, and HIF1A genes. To enable further analysis, we used standard inhibitors of *BIRC5* (flavokawain A; CID_5355469), *FLT4* (sorafenib; CID_216239), and HIF1A (belzutifan; CID_117947097), which were retrieved from PubChem as SDF files. The obtained SDF files were converted to PDB format using PyMol software. We obtained the crystal structures of *BIRC5* (1xox), *FLT4* (4bsk), and *HIF1A* (1l3e) from the Protein Data Bank in PDB format. The PDB files were then converted to PDBQT file format using autodock software [74], which enabled further processing. Docking simulations were performed using these 1.5.6) converted files. To analyze and visualize the docking results, we utilized BIOVIA discovery studio software [75].

## 3. Results

### 3.1. Identification of Differentially Expressed Genes (DEGs)

Four microarray datasets of NSCLC DEGs were retrieved from the GEO website, which we used to perform the gap analysis; we set the *p*-value as *p* < 0.05 and ∣log 2FC∣ > 1.5 as the statistical standards. To perform the analysis, from GSE2088, we sourced 48 tumor samples and 30 normal samples; from GSE6044, we sourced 9 tumor samples and 5 normal samples; from GSE19188, we sourced 91 tumor samples and 65 normal samples; and from GSE68465, we sourced 89 tumor samples and 67 normal samples. The expression data, which included upregulated and downregulated genes, were presented in volcano plots (Figure 1A–D).

### 3.2. Differential Expression of BIRC5/HIF1A/FLT4 in Normal, Tumor, and Metastatic NSCLC Samples

The number of overlapping upregulated genes from the GSE2088, GSE6044, GSE19188, and GSE68465 datasets was 6, including the following genes: *FLT4, NQO1*, *HIF1A*, *CXCL14*, *TFAP21*, and *BIRC5.* As displayed in the Venn diagram and heatmap in (Figure 2A,B), to validate and compare expression levels of *BIRC5/HIF1A/FLT4* in NSCLC, we used the TNM plot tool, and samples were obtained using RNA-Seq data. Based on these results, expression levels of *BIRC5, HIF1A*, and *FLT4* associated with primary NSCLC tumor progression and metastasis were compared to normal samples (Figure 2C–H). The Wilcoxon test was employed to assess the statistical significance of the differentially expressed genes (DEGs), with (*) *p* < 0.05 indicating significance.

### 3.3. Validation of Pathological Stages of the BIRC5/HIF1A/FLT4 Oncogenes in NSCLC and Their Prognoses

We further determined the pathological stage plots of BIRC5/HIF1A/FLT4 in NSCLC. Based on our findings, expression of *BIRC5*, *HIF1A*, and *FLT4* were significantly higher in stages 2, 3, and 4 than in stage 1. This result, thus, suggests that BIRC5/HIF1A/FLT4 signaling may promote tumor progression in NSCLC (Figure 3A–C). We further used UALCAN to analyze the expression of BIRC5/HIF1A/FLT4 in LUAD based on histological subtypes from TCGA samples. Interestingly, when BIRC5/HIF1A/FLT4 were upregulated in LUAD tissues, they exhibited a high presence in solid pattern-predominant adenocarcinomas, which are large and aggressive tumors with poor prognoses (Figure 3D–F) [76]. We also used Gepia2 to determine the prognostic significance of BIRC5/HIF1A/FLT4 expression in LUAD. As anticipated, results showed that high expression of all of these genes in LUAD were associated with shorter OS, and the hazards ratio (HR) was calculated based on the Cox PH Model, with 95% CI set as the dotted line and Cutoff values (high and low) set at 50%; *p* < 0.05 was considered statistically significant (Figure 3G–I).

### 3.4. PPI Network Construction and the GEA

Protein interactions were analyzed using the STRING database. A confidence score higher than 0.9 was deemed to indicate the most significant interactions, and the network was further constructed using 7 nodes, 21 edges, an average node degree of 6, an average local clustering coefficient of 1, the number of edges being expanded to 6, and a PPI enrichment *p* value of 3.51 × 10^−6^. Active interactions were determined through various sources, including text mining, experimental data, databases, coexpression patterns, spatial proximity, gene fusion events, and co-occurrence analysis (Figure 4A). For further exploration, we used enriched PPI clustering networks based on STRING results to perform a GEA with the DAVID database, and we further utilized Funrich. Functional enrichments included gene ontology (GO) that involved biological processes such as anti-apoptosis, immune response, regulation of gene expression and epigenetics, morphogenesis, cell migration, protein metabolism, regulation of nucleobases, cell communication, and signaling transduction (Figure 4B). Affected biological (Kyoto Encyclopedia of Genes and Genomes; KEGG) pathways included the FOXM1 transcription factor network, notch signaling pathway, HIF-1α transcription factor, and FLT4 signaling network (Figure 4C). We also used a network analysis, and KEGG pathway enrichment showed co-expression of the BIRC5/HIF1A/FLT4. Oncogenes within the same network cluster were analyzed based on their network topology using the Igraph R package, and the results were visualized using a force atlas layout (Figure 4D). Statistical significance was set at a threshold of *p* < 0.05.

### 3.5. Analysis of BIRC5/HIF1A/FLT4 Genetic Mutations in LUAD Solid Tumors

The genetic alterations and gene expression changes in the BIRC5, HIF1A, and FLT4 oncogenes in lung adenocarcinoma (LUAD) were analyzed using muTarget software. The top two highly expressed genes linked to *BIRC5* were *TP53* and *TTN*; for *FLT4,* they were *CCDC129* and *LTN1*; and for *HIF1A,* they were *PLOR2A* and *POTEG* compared to the wild type, and all of these genes were associated with unfavorable prognoses (Figure 5A–F). Moreover, we utilized the oncoprint webtool, which is embedded in cBioPortal software, to further analyze genetic alterations and CNVs of BIRC5/HIF1A/FLT4 in LUAD. The analysis revealed the following percentages of gene amplification: 2.3% for BIRC5, 1.9% for HIF1A, and 1.7% and 1.9% for FLT4 in LUAD. Gene change categories included missense mutations, amplifications, deep deletions, and no alterations, which are denoted by green, red, blue, and grey, respectively. (Figure 5G). We further analyzed the alteration frequencies of the *BIRC5* and *FLT4* oncogenes, as shown in the bar graphs (Figure 5H,I), with *p* < 0.001 considered to be significant.

### 3.6. Single-Cell RNA Sequencing (scRNA-Seq) Profiling Unveiled the High Abundance and Immunosuppressive Role of BIRC5/HIF1A/FLT4 within the TME of Both Primary and Metastatic Non-Small Cell Lung Cancers (NSCLC)

We explored the TISCH scRNA-Seq database of NSCLC from the GSE148071 dataset, which comprised samples from 12 patients diagnosed with primary and metastatic NSCLC. The detailed annotation of cell types at the single-cell level, which facilitated our investigation of the tumor microenvironment (TME); accordingly, we found abundances of major linear cell types within the NSCLC TME, and these cell types included malignant cells, fibroblasts, epithelial cells, plasma, CD8 T cells, T proliferation cells, endothelial cells, basal cells, alveolar cells, and mono cells (Figure 6A). The analysis of differential gene expression in the scRNA-Seq data revealed that *BIRC5*, *HIF1A*, and *FLT4* were overexpressed in malignant tissues [67]. By conducting a meta-analysis of differentially expressed genes (DEGs) within each cell type of the tumor microenvironment (TME), we observed that increased expression levels of specific genes, particularly *BIRC5* and *HIF1A*, occurred in epithelial cells, CD8 T cells, T proliferative cells, mono/macro cells, basal cells, fibroblasts, and malignant cells (Figure 6B,C), while high expression levels of *FLT4* occurred in malignant cells, CD8 T cells, mono/macro cells, and basal cells (Figure 6D).

### 3.7. Correlations between BIRC5/HIF1A/FLT4 and Infiltrating Immune Cells in NSCLC Patients

The TME plays a crucial role in cancer initiation and progression. However, the association between the TME and tumor prognosis remains elusive. Herein, we utilized a web-based program of a tumor-infiltrating immune cell algorithm (TIMER 2.0) to explore how our target gene correlates with the TME in NSCLC. Accordingly, we used a TIMER database analysis to determine correlations between *BIRC5, HIF1A,* and *TLF4* and infiltrating immune cells. In order to understand the connections between BIRC5, HIF1A, and FLT4 expression and specific immune cells, we conducted a correlation analysis between these oncogenes and markers of cancer-associated fibroblasts (CAFs), considering the influence of sample purity. As expected, results showed correlations between BIRC5/HIF1A/FLT4 and CAFs in NSCLC (Figure 7A–C). In addition, we analyzed mutations of BIRC5/HIF1A/FLT4 in CAFs using the mutation module from TCGA clinical outcomes in the TIMER algorithm (Figure 7D–F). Moreover, the expression of BIRC5, HIF1A, and FLT4 were analyzed across various immune subtypes, namely C1 (associated with wound healing), C2 (dominated by IFN-γ), C3 (inflammatory), C4 (characterized by lymphocyte depletion), C5 (immunologically quiet), and C6 (dominated by TGF-β). Interestingly, the BIRC5/HIF1A/FLT4 oncogenes were highly expressed in all of the above-mentioned immune subtypes, except for the wound-healing subtype (Figure 7G–I).

### 3.8. Digital Flow Cytometric Analysis of NSCLC RNA-Seq Bulk Tumors Revealed Abundant Infiltrating Immune Cells Associated with Poor Clinical Outcomes

To profile bulk RNA-Seq expression of tumor-infiltrating cells, we used CIBERSORTx, which is an online and versatile analytical tool that allows a signature to be extracted from single-cell RNA-Seq data, as well as quantification of cell fractions from differential gene expression of bulk tumors. For each sample size, a subset of tumors was randomly selected from a larger cohort (n = 302) in 10 iterations. The results were displayed with and without adaptive noise filtration. The data were presented using boxplots, where the center line represents the median, the box limits indicate the upper and lower quartiles, the whiskers extend to 1.5 times the interquartile range, and any points beyond the whiskers represent outliers (Figure 8A). Heatmaps were utilized to compare the expression profiles of imputed and ground truth data for immune (CD45^+^), epithelial/cancer (EpCAM), and stromal (CD10^+^ and CD34^+^) subsets. Genes that were either not predicted to be expressed or were eliminated through adaptive noise filtration were indicated by items that were navy blue in color. Figure 8B–E show positive correlations between CD10, EpCAM, and CD45 in NSCLC, which were reported to be associated with poor clinical outcomes [77]. We also performed an OS analysis plot from Gepia2, and we found that expression of CD10, EpCAM, and CD45 in NSCLC were, indeed, correlated with poor prognoses (Figure 8F–H).

### 3.9. Rationale for Drug Design via Scaffold Hopping to Determine the Physicochemical Properties of the Bioactive Compound—HN-N 07—And Its Anticancer Activities against NSCLC Cell Lines

Pharmacophore hybridization and the exploration of different bioactive compound scaffolds are valuable strategies in the design and development of new drugs [63]. In the present study, we used a quinoline to synthesize our small molecule—HN-N07. Quinolines and their derivatives play multiple roles due to their biological activities, such as anti-inflammation and anticancer immunomodulation [55,56,57,58]. Furthermore, various anticancer drugs, such as irinotecan and topotecan, incorporate quinolones as their primary structural framework, which contribute to their therapeutic properties [78] (Figure 9A). We evaluated the impact of HN-N07 on the growth and viability of NCI-60 NSCLC cell lines and observed significant antiproliferative and cytotoxic effects. HN-N07 demonstrated potent anticancer properties against these specific cancer cell lines. The antiproliferative activities of HN-N07 were shown after an initial single dose (10 μM), through which the effects of the compound are represented based on the percentage growth modified by the treatment as follows: A549/ATCC (23.38%), (HOP-62 56.42%), HOP-92 (−10.12%), NCI-H226 (58.08%), NCI-H23 (72.59%), NCI-H322M (49.16%), NCI-H460 (−71.14%), and NCI-H522 (−8.99%) (Figure 9B). Since HN-N07 displayed antiproliferative activities against NSCLC cell lines screened from the NCI database at an initial single dose of 10 μM, we further investigated the effect of the compound when administered in a dose-dependent manner. The 50% growth inhibition (GI50) concentrations of HN-N07 against NSCLC cell lines were in the range 2.92~4.38 μM. The most sensitive cell line was HOP-62 (2.92 μM), followed by NCI-H522 (3.07 μM), A549/ATCC (3.17 μM), and NCI-H322M (3.18 μM), while the lowest GI activity was shown with NCI-H460 (3.38 μM) and NCI-H226 cells (4.38 μM)). A growth percentage value of 100 indicates the growth of untreated cells, whereas a value of 0 indicates no overall growth during the experimental period. A value of −100 signifies complete cell death by the end of the experiment (Figure 9C). To further explore this topic, we investigated druggable target genes for HN-N07. Interestingly, we identified several targets, which included kinases, family A–G protein-coupled receptors, proteases, and transcription factors (Figure 9D).

### 3.10. BIRC5/HIF1A/FLT4 Are Potential Target Genes for the HN-N07 Compound

We used the Swisstarget prediction tool to investigate druggable target genes for HN-N07. Interestingly, we identified several targets which included kinases, family A–G protein-coupled receptors, proteases, and transcription factors. The HN-N07 small molecule was also shown to target oncogenes, including *BIRC5*, *FLT4*, *HIF1A*, dopamine receptor D2 (*DRD2*), mammalian target of rapamycin (*mTOR*), dipeptidyl peptidase-4 inhibitor (*DPP-4*), proto-oncogene B-Raf (*BRAF*), and others, as shown in Table 1.

### 3.11. Molecular Docking Revealed Putative Interactions of HN-N07 with the BIRC5, FLT4, and HIFA Oncogenes

Our computational molecular docking analysis uncovered potential binding capabilities of HN-N07 with the *BIRC5, FLT4*, and *HIFA* oncogenes. Accordingly, after docking our compound with these individual genes, we obtained the following the Gibbs free binding energy (ΔG) results: *BIRC5* (PDB: 1xox = −8.2 kcal/mol); *FLT4* (PDB: 4bsk = −7.7 kcal/mol); and *HIF1A* (PDB: 1l3e = −8.2 kcal/mol) (Figure 10A,B, Figure 11A,B and Figure 12A,B). The results of the docking analysis were visualized using Discovery Studio, demonstrating interactions that involved conventional hydrogen bonds and their corresponding distance constraints. In a further analysis, we performed molecular docking with standard inhibitors of *BIRC5* (flavokawain A), *FLT4* (sorafenib), and *HIF1A* (belzutifan), and compared those results to our HN-N07 small molecule. Interestingly, our compound had higher binding affinities than flavokawain A (−6.8 kcal/mol) and belzutifan (−8.2 kcal/mol). However, sorafenib exhibited a much higher binding energy of −7.9 kcal/mol (Figure 10C,D, Figure 11C,D and Figure 12C,D), which was slightly higher than that of HN-N07 (−7.6 kcal/mol). Collectively, these findings suggest that HN-N07 may be a potential inhibitor of the *BIRC5/HIF1A/FLT4* oncogenic signaling pathway in NSCLC (Table 2, Table 3 and Table 4).

## 4. Discussion

Despite the advanced optional treatment modalities for NSCLC, including surgery, radiation therapy, and chemotherapy, which depend on various factors, such as cancer types and histological subtypes, that make treatments less effective, OS is still approximately <5 years [7,79,80,81,82]. Improved insights into biological pathways have shed light on the development of targeted therapies and antiangiogenic drugs, which has significantly improved the survival of patients [83,84].

Angiogenic pathways are essential targets in the molecular regulation of the NSCLC tumor microenvironment (TME), influencing tumor progression and metastasis [85]. Vascular endothelial growth factor (VEGF) is a key factor involved in angiogenesis and highly expressed in various tumors, including NSCLC [86]. Bevacizumab, which is a monoclonal antibody targeting circulating VEGF, was the first angiogenesis inhibitor approved for treatment of advanced NSCLC, though its use is limited to non-squamous histology in first-line treatment. Currently, bevacizumab, in combination with platinum-based chemotherapy, is the only approved treatment for advanced NSCLC in the first-line setting. Ongoing clinical investigations are evaluating other antiangiogenic agents, such as sorafenib and sunitinib [85,87]. Accumulating studies have shown that *BIRC5,* which are also known as survivin, is an immune-associated gene, which has also been shown to regulate metastasis and angiogenesis in tumors and is highly expressed in NSCLC [88]. The co-expression of *BIRC5* and survivin was shown in NSCLC, thus illustrating its great potential as a therapeutic target for treatment development [89]. Moreover, studies showed that angiogenesis is prompted by hypoxia as a result of insufficient new blood vessels [90]. Additionally, hypoxia was found to activate HIF-1α within the NSCLC TME, leading to VEGF expression [50,91,92,93], which, in turn, results in distant metastasis and poor prognoses [51]. Additionally, the presence of hypoxia within the tumor microenvironment has an impact on both the early and late stages of the disease [94,95].

In the present study, we have analyzed a sample of patients affected by metastatic disease and compared them to normal samples using bioinformatics analysis of non-small cell lung cancer patients. We used the TNM plot tool, and samples were obtained using RNA-Seq data. Our results showed that high expression levels of *BIRC5*, *FLT*, and *HIF1A* were more clearly associated with primary NSCLC tumor progression and metastasis than normal samples. These findings are in line with the study conducted by Aldo et al. in 2019. Interestingly, based on results of the correlation analysis, all of the oncogenic signatures were also coexpressed in NSCLC. Since current treatment is influenced by factors such as cancer types and histological subtypes, we determined the prognostic relevance of *BIRC5/FLT4/HIF1A* at different stages of LUAD, and we found that *BIRC5*, *FLT4,* and *HIF1A* were higher in stages 2, 3, and 4 than in stage 1, where they were significantly lower. This result suggests that *BIRC5/HIF1A/FLT4* signaling may promote tumor progression in NSCLC. The expression of this signature in LUAD based on histological subtypes from TCGA samples exhibited a high presence of a solid pattern-predominant adenocarcinoma, which is an aggressive large tumor associated with poor clinical outcomes.

The TME plays crucial roles in cancer initiation and progression. However, the association between the TME and tumor prognosis remains elusive. Herein, we utilized a web-based program with a tumor-infiltrating immune cell algorithm (TIMER 2.0) to explore correlations between our target genes and the TME in NSCLC. In order to determine the associations between BIRC5, HIF1A, and FLT4 expression and specific immune cells, we conducted a correlation analysis that accounted for the purity-adjusted CAF markers. As anticipated, results showed correlations between *BIRC5/HIF1A/FLT4* and CAFs in NSCLC. Moreover, we exploited a CIBERSORTx digital flow cytometric analysis of NSCLC RNA-Seq bulk tumors, and we identified that immune (CD45^+^), epithelial/cancer (EpCAM), and stromal (CD10^+^ and CD34^+^) subsets were expressed in NSCLC and associated with poor prognoses. Due to their broad efficacy and safety compared to traditional chemotherapeutic regimens, small molecules, which are kinase inhibitors, have become the most recognized cancer treatments [54,96]. Herein, we evaluated the potential inhibitory activities of HN-N07, which is a quinoline-derived small molecule derived in our lab. Accordingly, we performed computational molecular docking to determine ligand–receptor interactions. Docking results between HN-N07 and the *BIRC5*, *FLT4,* and *HIF1A* oncogenes revealed unique binding energies. These energies were significantly higher than the standard inhibitors of flavokawain-A and belzutifan. However, sorafenib exhibited a binding energy that was slightly higher than that of HN-N07. Collectively, these findings suggest that HN-N07 may be a potentially inhibitor of the oncogenic signaling pathway in NSCLC.

## 5. Conclusions

In conclusion, we identified the *BIRC5/HIF1A/FLT4* signature as a targetable signature correlated with angiogenic pathways in NSCLC. We used a computational analysis and identified the *BIRC5/HIF1A/FLT4* oncogenes as being highly upregulated in NSCLC and associated with cancer progression and poor prognoses. Docking results of HN-N07 with the *BIRC5*, *FLT4*, and *HIF1A* oncogenes revealed unique binding energies of −8.2, −7.7, and −8.2 kcal/mol. These results were significantly higher than those of two standard inhibitors. Collectively, these findings suggest that HN-N07 may be a potential inhibitor of an oncogenic signaling pathway in NSCLC. Currently, further in vitro and in vivo investigations in tumor-bearing mice are in progress to study the potential treatment efficacies of the novel HN-N07 small molecule.

## Figures and Tables

**Figure 1 biomedicines-11-02011-f001:**
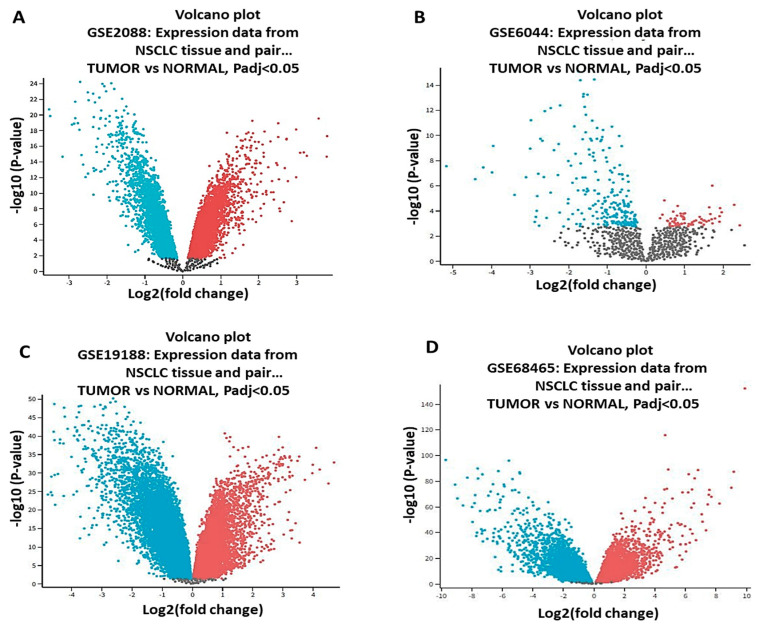
Volcano plots that depict the differential expression of genes in the following datasets: (**A**) GSE2088, (**B**) GSE6044, (**C**) GSE19188, and (**D**) GSE68465. All downregulated genes are represented by blue color, while the upregulated genes are shown in red color, with *p*< 0.05 considered statistically significant.

**Figure 2 biomedicines-11-02011-f002:**
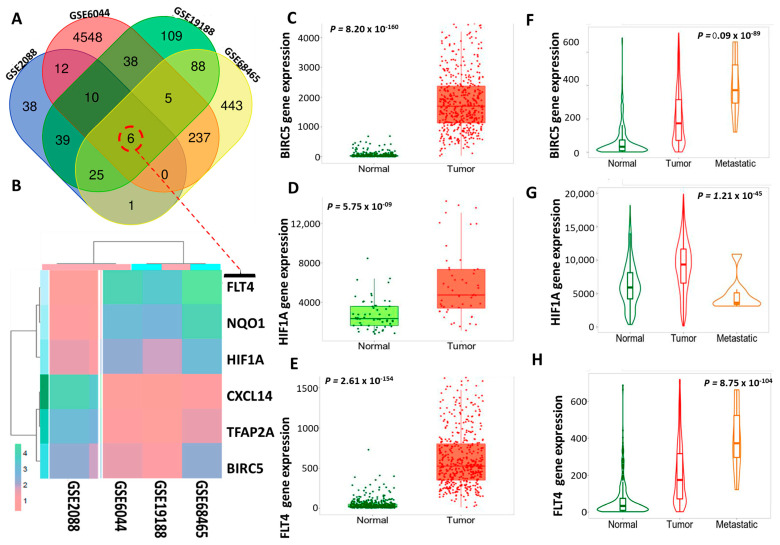
Differential expression of BIRC5/HIFA1/FLT4 in normal, tumor, and metastatic samples of non-small cell lung cancer (NSCLC). (**A**) Venn diagram that shows 6 overlapping upregulated genes in NSCLC samples. (**B**) Heat maps of the overlapping upregulated DEGs obtained using the four datasets. Overexpression of BIRC5/HIFA1/FLT4 in NSCLC tumor cells compared to the adjacent normal tissues (**C**–**H**). Dysregulation of BIRC5/HIFA1/FLT4 in tumor and metastatic NSCLC tumor tissues compared to normal tissues. The statistical significance of the differentially expressed genes was determined using the Wilcoxon test, with a significance threshold being set at *p* < 0.05.

**Figure 3 biomedicines-11-02011-f003:**
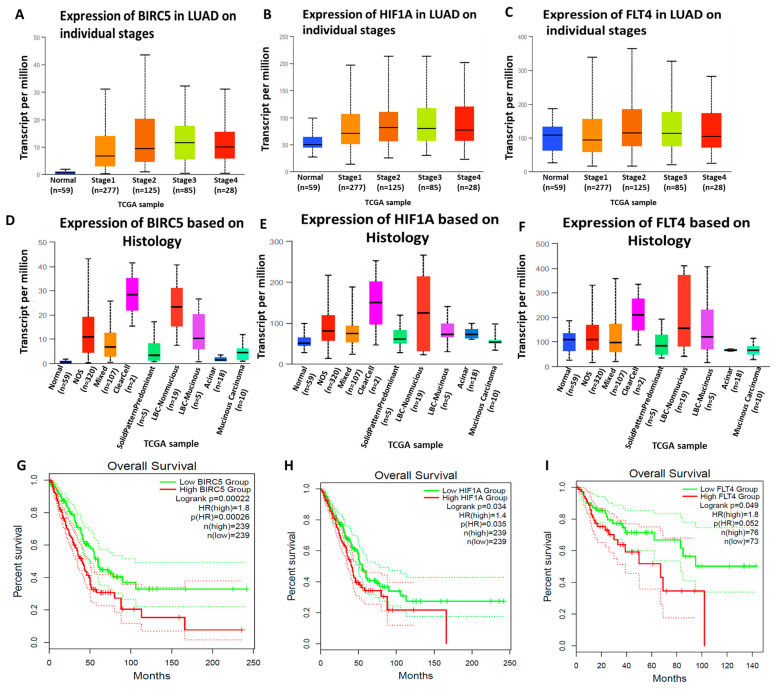
The BIRC5/HIF1A/FLT4 gene signature was overexpressed in NSCLC and linked to its progression. (**A**–**C**) BIRC5/HIF1A/FLT4 were more highly expressed in stages II to IV of NSCLC than in stage I. Transcript levels of BIRC5/HIF1A/FLT4 (**D**–**F**). Histological subtype analysis showing that if upregulated in LUAD tissues, BIRC5/HIF1A/FLT4 exhibited a high presence in larger aggressive tumors with poor prognoses. (**G**–**I**) The Kaplan–Meier plot indicated that the increased expression of BIRC5/HIF1A/FLT4 was correlated with a decrease in overall survival, with statistical significance set at *p* < 0.05.

**Figure 4 biomedicines-11-02011-f004:**
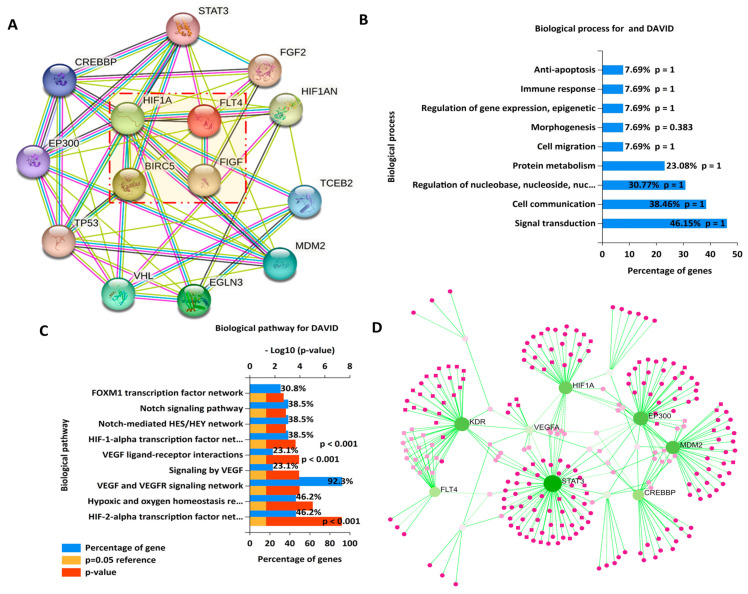
The protein–protein interaction (PPI) network demonstrated the interactions between the BIRC5, HIF1A, and FLT4 oncogenes in NSCLC. (**A**) The clustering network consisted of 7 nodes, 21 edges, an average node degree of 6, an average local clustering coefficient of 1, the number of edges being expanded to 6, and a PPI enrichment *p* value of 3.51 × 10^−6^. Active interactions were based on text mining, experiments, databases, coexpression, neighborhoods, gene fusion. and co-occurrence. *p* < 0.05 was considered to be statistically significant. (**B**) Top biological processes (BPs), (**C**) KEGG pathways, and (**D**) signaling network analysis were sourced from the KEGG database, with coexpression of the BIRC5/HIF1A/FLT4 oncogenes displayed, and the criterion was set to *p* < 0.05 in each panel.

**Figure 5 biomedicines-11-02011-f005:**
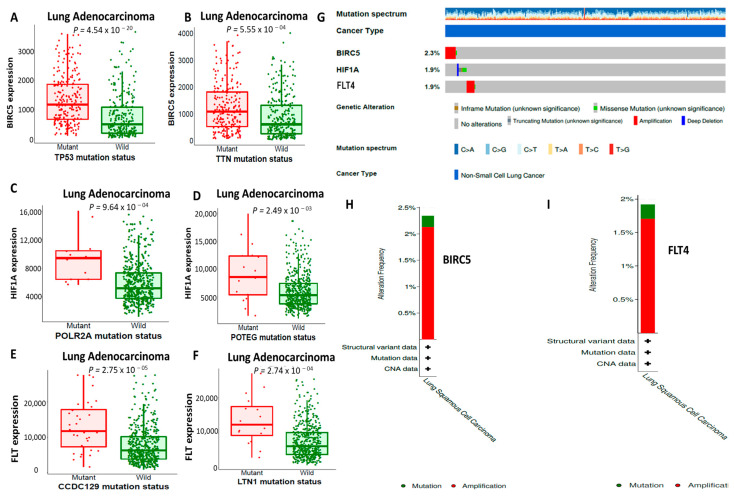
BIRC5/HIF1A/FLT4 mutations were associated with worse prognosis in NSCLC. (**A**–**F**) Top two highly expressed genes linked to *BIRC5* were *TP53* and *TTN*; for *FLT4,* they were *CCDC129* and *LTN1*; and for *HIF1A,* they were *PLOR2A* and *POTEG* compared to the wild type. (**G**) Genetic alterations and copy number variations (CNVs) in BIRC5/HIF1A/FLT4 in lung adenocarcinoma (LUAD) were based on percentages of separate genes due to amplification (**H**,**I**). Bar graphs that show alteration frequencies of the *BIRC5* and *FLT4* oncogenes.

**Figure 6 biomedicines-11-02011-f006:**
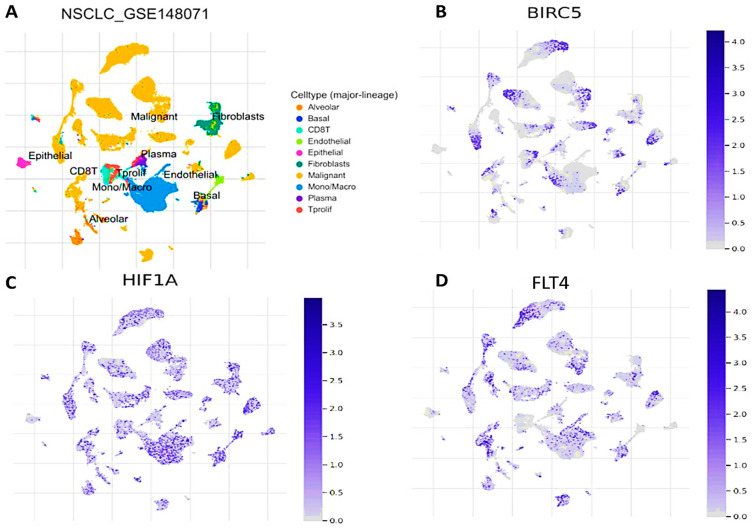
Cell sequencing leveraging of BIRC5/HIF1A/FLT4 in the tumor microenvironment of primary and metastatic NSCLC. (**A**) Cell-type distributions sourced from the NSCLC GSE148071 dataset The single-cell resolution analysis of the GSE148071 dataset in the TISCH database allowed us to examine the expression patterns of BIRC5 (**B**), HIF1A (**C**), and FLT4 (**D**) in various cell types.

**Figure 7 biomedicines-11-02011-f007:**
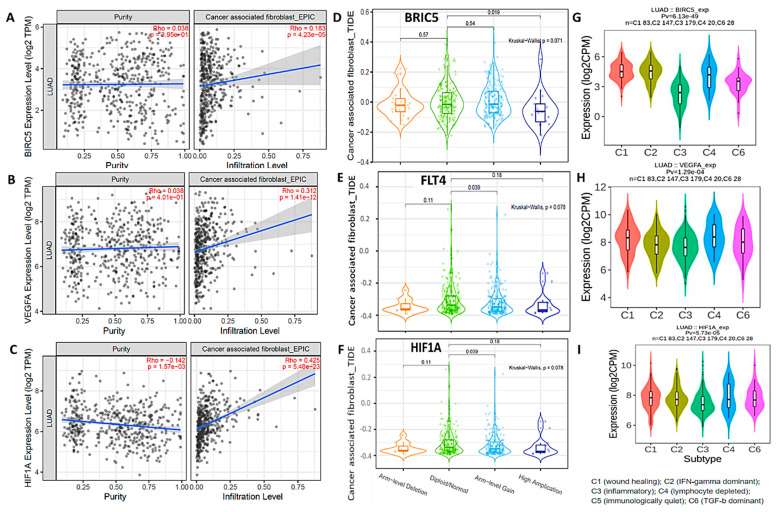
Correlations between BIRC5/HIF1A/FLT4 and infiltrating immune cells in NSCLC patients. (**A**–**C**) *BIRC5/HIF1A/FLT4* expression levels were significantly correlated with cancer-associated fibroblast (CAF) infiltration in NSCLC. (**D**–**F**) Mutation analysis of BIRC5/HIF1A/FLT4 in CAFs. (**G**–**I**) *BIRC5/HIF1A/FLT4* were more highly expressed in different immune subtypes.

**Figure 8 biomedicines-11-02011-f008:**
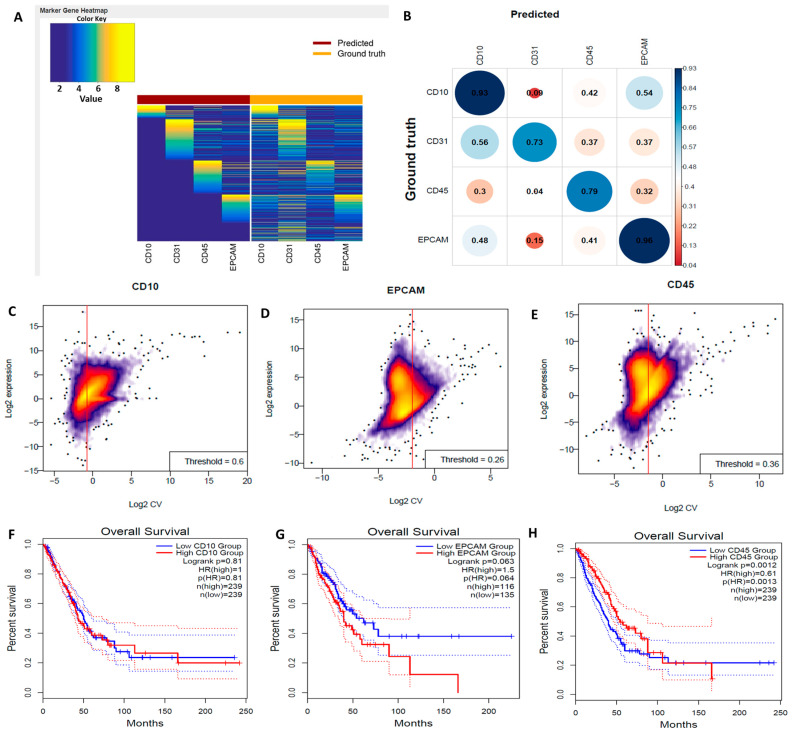
In silico flow cytometric analysis of non-small cell lung cancer (NSCLC) RNA-Seq bulk tumors revealed abundant infiltrating immune cells that were associated with poor clinical outcomes. (**A**) Heat map that compares imputed and ground truth expression profiles for immune (CD45^+^), epithelial/cancer (EpCAM), and stromal (CD10^+^ and CD34^+^) subsets. (**B**–**E**) Positive correlations of CD10, EpCAM, and CD45 in NSCLC. (**F**–**H**) Overall survival analysis revealed that expression of CD10, EpCAM, and CD45 in NSCLC were correlated with a poor prognosis, with *p* < 0.05 considered significant.

**Figure 9 biomedicines-11-02011-f009:**
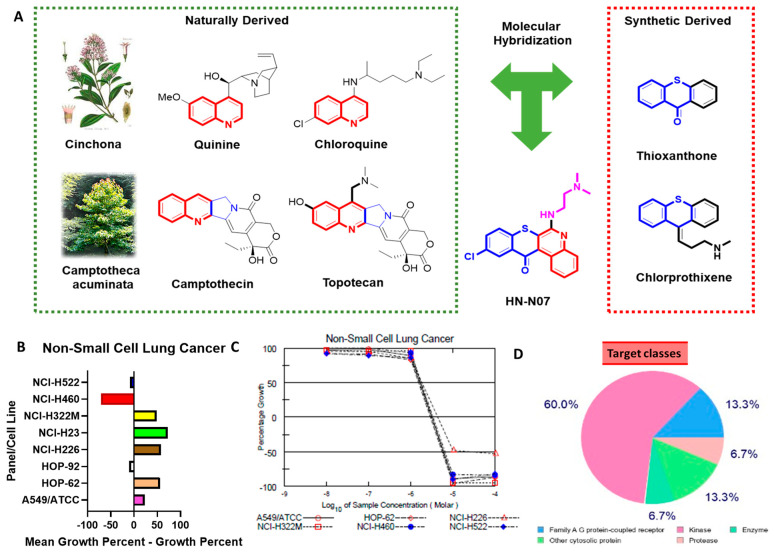
(**A**–**D**) Rationale for drug design via scaffold hopping for physicochemical properties of the bioactive compound HN-N07, as well as its anticancer activities against non-small cell lung cancer (NSCLC) cell lines.

**Figure 10 biomedicines-11-02011-f010:**
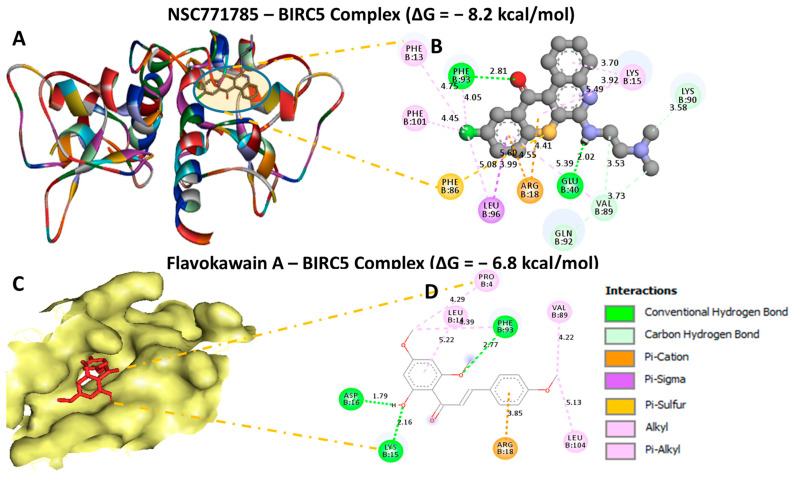
Docking profiles of *BIRC5* with HN-N07 and flavokawain A, which is its standard inhibitor. (**A**,**C**) Three-dimensional (3D) representations of the ligand–receptor complex that show respective binding energies of −8.2 and −6.8 kcal/mol (**B**,**D**) for *BIRC5* in complex with HN-N07 and flavokawain A. Interacting amino acid residues and types of interactions that occur between the ligands.

**Figure 11 biomedicines-11-02011-f011:**
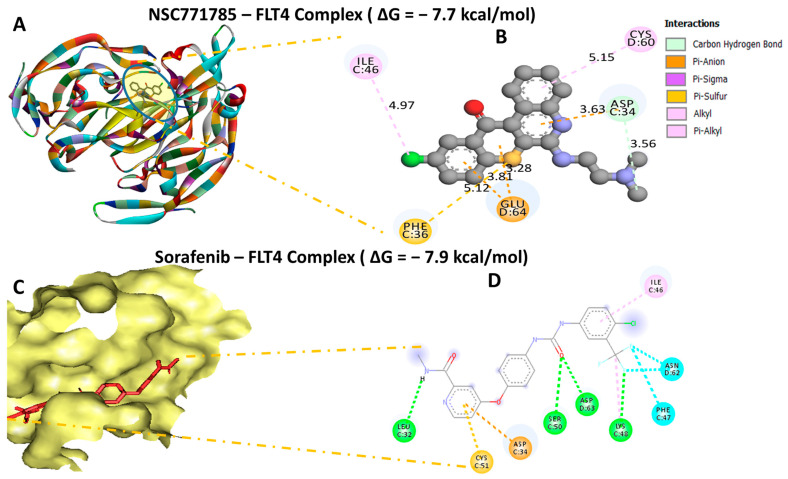
Docking profiles of *FLT4* with HN-N07 and sorafenib, which is its standard inhibitor. (**A**,**C**) Three-dimensional (3D) representations of the ligand–receptor complex that show respective binding energies of −7.7 and −7.9 kcal/mol (**B**,**D**) for *FLT4* in complex with HN-N07 and sorafenib. (**B**) Interacting amino acid residues and types of interactions that occur between the ligands.

**Figure 12 biomedicines-11-02011-f012:**
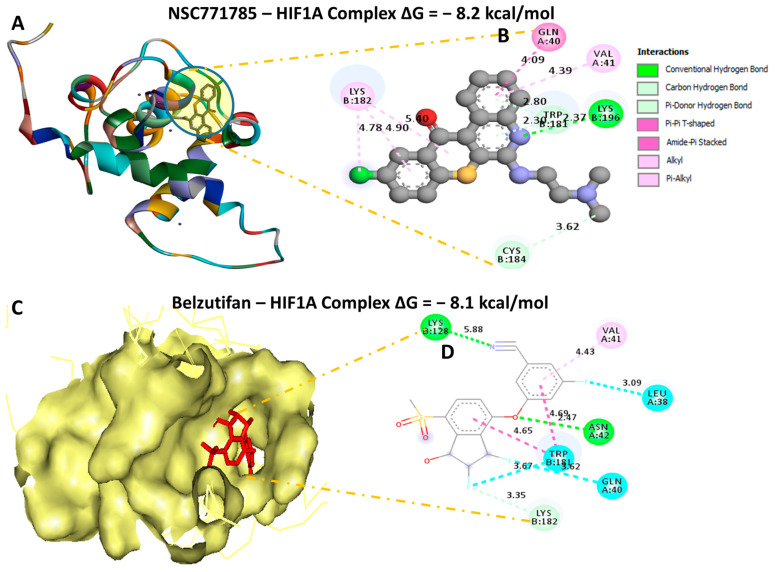
Docking profiles of *HIF1A* with HN-N07 and belzutifan, which is its standard inhibitor. (**A**,**C**) Three-dimensional (3D) representations of the ligand–receptor complex that shows respective binding energies of −8.1 (**B**,**D**) and −8.2 kcal/mol for HIF1A in complex with HN-N07 and belzutifan. (**B**) Interacting amino acid residues and types of interactions that occur between the ligands.

**Table 1 biomedicines-11-02011-t001:** Protein targets of NSC77178 that show common names, Uniprot and ChEMBL IDs, and target classes of specific compounds.

Target	Common Name	Uniprot ID	ChEMBL ID	Target Class
Serotonin 1f (5-HT1f) receptor	**HTR1F**	P30939	CHEMBL1805	Family A–G protein-coupled receptor
Dopamine D2 receptor (by homology)	**DRD2**	P14416	CHEMBL217	Family A–G protein-coupled receptor
Vascular endothelial growth factor receptor 3	**FLT4**	P35916	CHEMBL1955	Kinase
Serine/threonine-protein kinase mTOR	**MTOR**	P42345	CHEMBL2842	Kinase
Cyclin-dependent kinase 2/cyclin A	**CDK2 CCNA1 CCNA2**	P24941 P78396 P20248	CHEMBL2094128	Other cytosolic protein
Dipeptidyl peptidase IV	**DPP4**	P27487	CHEMBL284	Protease
Serine/threonine-protein kinase B-raf	**BRAF**	P15056	CHEMBL5145	Kinase
Dual specificity mitogen-activated protein kinase 1	**MAP2K1**	Q02750	CHEMBL3587	Kinase
Fibroblast growth factor receptor 1	**FGFR1**	P11362	CHEMBL3650	Kinase
Cyclin-dependent kinase 4	**CDK4**	P11802	CHEMBL331	Kinase
Gonadotropin-releasing hormone receptor	**GNRHR**	P30968	CHEMBL1855	Family A–G protein-coupled receptor
Hypoxia-inducible factor 1 alpha	**HIF1A**	Q16665	CHEMBL4261	Transcription factor
Hepatocyte growth factor receptor	**MET**	P08581	CHEMBL3717	Kinase
Peptide N-myristoyltransferase 1	**NMT1**	P30419	CHEMBL2593	Enzyme

**Table 2 biomedicines-11-02011-t002:** Comparative docking profiles of HN-N07 against BIRC5 and its standard drug, which is flavokawain A.

HN-N07—BIRC5 Complex ΔG = −8.2 kcal/mol		Flavokawain A—BIRC5 Complex ΔG = −6.8	
Type of Interactions and Number of Bonds	Distance of Interacting Amino Acids	Type of Interactions and Number of Bonds	Distance of Interacting Amino Acids
Conventional hydrogen bond (3)	PHE93 (2.81 Å), GLU40 (2.02 Å)	Conventional hydrogen bond (3)	PHE93 (2.77 Å), ASP16 (1.79 Å), LYS15 (2.16 Å),
Carbon hydrogen bond	GLN92, VAL89, LYS91	Pi-Cation	ARG18
Pi-Cation	ARG18	Alkyl	LEU14, PRO4
Pi-sigma	LEU96	Pi-Alkyl	VAL89, LEU104
Pi-Sulfur	PHE86		
Pi-Alkyl	PHE13, PHE101, LYS15		

**Table 3 biomedicines-11-02011-t003:** Comparative docking profiles of HN-N07 against FLT4 and its standard drug, which is sorafenib.

HN-N07—FLT4 Complex (ΔG = −7.7 kcal/mol)		Sorafenib—FLT4 Complex (ΔG = −7.9 kcal/mol)	
Type of Interactions and Number of Bonds	Distance of Interacting Amino Acids	Type of Interactions and Number of Bonds	Distance of Interacting Amino Acids
Carbon hydrogen bond	ASP34	Conventional hydrogen bond (4)	LEU32 (2.85 Å), SER50 (1.79 Å), ASP63 (2.16 Å), LYS48 (2.69 Å)
Pi-Anion	GLU64	Halogen	ASN62, PHE47
Pi-Sigma	CYS60	PI-Anion	ASN34
Pi-Sulfur	PHE35	Pi-Sulfur	CYS51
Alkyl	ILE46	Alkyl	ILE46

**Table 4 biomedicines-11-02011-t004:** Comparative docking profiles of HN-N07 against HIF1A and its standard drug, which is belzutifan.

HN-N07—HIF1A Complex ΔG = −8.2 kcal/mol		Belzutifan—HIF1A Complex ΔG = −8.1 kcal/mol	
Type of Interactions and Number of Bonds	Distance of Interacting Amino Acids	Type of Interactions and Number of Bonds	Distance of Interacting Amino Acids
Conventional hydrogen bond (2)	LYS196 (2.37 Å)	Conventional hydrogen bond (2)	LEU128 (5.88 Å), ASN42 (2.47 Å)
Carbon hydrogen bond	CYS184	Carbon hydrogen bond	LYS182
Pi-Donor Hydrogen Bond	TRP181	Halogen	TRP181, GLN40, LEU38
Pi-Pi-T-shaped	GLN40	Pi-Alkyl	VAL41
Alkyl	LYS182		
Pi-Alkyl	VAL41		

## Data Availability

The data supporting the findings of this study will be made available in a transparent and accessible manner, without any unnecessary restrictions.

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
