# Peer review of "In Silico Evaluation of HN-N07 Small Molecule as an Inhibitor of Angiogenesis and Lymphangiogenesis Oncogenic Signatures in Non-Small Cell Lung Cancer"

_biomedicines, 2023, doi:10.3390/biomedicines11072011_

Round 1

Reviewer 1 Report

The authors conducted an analytical computational study to investigate a new molecule and its corresponding pathway. All experiments involved in silico studies, except for one in vitro cytotoxicity experiment using cancer cell lines.

I have the following comments:

-Please rephrase the title to accurately reflect the study's focus as in its current form, it is not understandable.

-The authors' approach of using in silico methods is commendable as it allows for cost-effective and efficient exploration of potential interactions and properties. The inclusion of an in vitro cytotoxicity experiment is essential for validating the potential anti-cancer properties of the studied molecule. However, the authors should consider conducting additional in vitro experiments to strengthen the significance of their findings in the future.

-In the discussion section, the authors could elaborate more on how their results compare with existing literature on similar molecules or pathways. This could help place their findings in a broader context and provide valuable insights for future research. It would also be helpful if the authors discussed any potential limitations or challenges faced during their computational analysis, as this can influence the interpretation of results.

-Moreover, it is important to consider the vast number of images and potentially relocate some to supplementary material.

-The authors should examine the molecule's toxicity in healthy cells alongside cancer cell lines, which is crucial in assessing its potential for clinical use. If the molecule exhibits significant toxicity towards normal cells, it may not be a viable candidate for therapeutic application in patients.

Overall, the study appears promising and contributes valuable information to the field of computer-aided drug discovery. Addressing the above points would further enhance the quality and impact of their research.

Moderate editing of English language required

Reviewer 2 Report

It is interestingy

you need to specify which technique was used to detect the factors BIRC5, FLT4, and HIF1A. Were proteins detected besides genes?

What test was used to determine survival? it must appear in the text

Is it possible to derive the general characteristics of the sample of patients included in the analysis?

for example, it would be interesting to know what interaction there is between transcriptional factors and exposure to tobacco smoke

Why not use the regression test?

explains why the Kruskal-Wallis test was used.

I suggest to include the followingr references useful for discussion

about hypoxia activated transcription factors involved in lung cancer metastases mechanisms

- Bioorg Med Chem. 2023 Jun 20;91:117384. 

- Oncotarget. 2019 Dec 17;10(66):7071-7079. 

Reviewer 3 Report

I have reviewed the paper entitled “In silico evaluation of HN-N07 small molecule as an inhibitor 2 angiogenesis and lymphangiogenesis oncogenic signatures in 3 non-small cell lung cancer”.  The detailed suggestions are listed below;

Abstract

The abstract of this research paper appears to be well-structured and informative, providing a clear overview of the study's objectives, methods, and findings. 

Introduction

The introduction of this research paper is well-structured and provides a good overview of the problem being addressed. It highlights the significant impact of NSCLC and the limitations of current treatment options, emphasizing the urgent need for novel drug targets. The introduction also provides a good overview of the role of BIRC5 in NSCLC and its association with angiogenesis and inhibition of cell apoptosis.

However, there are some areas where the introduction could be improved;

1.     The introduction could benefit from a clearer research question or hypothesis that the study aimed to address. While the introduction provides a good overview of the role of BIRC5, HIF1α, and FLT4 in NSCLC, it is not clear what specific question the study aimed to answer or how the research findings contribute to our understanding of the disease.

2.     the introduction could benefit from a more detailed discussion of the implications of the findings. While the introduction mentions that HN-N07 could be a potential inhibitor of oncogenic signaling pathways in NSCLC, it does not provide a clear indication of how this may impact the development of new therapies or contribute to our understanding of the disease.

Methods and Materials 

he section is well-organized and provides sufficient information for the reader to understand the methods used in the study. However, there are some areas that could benefit from further clarification or elaboration.

1.     In section 2.1, the authors describe the differential expression analysis of BIRC5/HIF1Α/FLT4 in normal, tumor, and metastatic tissues using the TNMplot and GEPIA2 online tools. The methods used for RNA-Seq profiling and statistical analysis are briefly mentioned, but more details would be helpful, such as the specific RNA-Seq library preparation and sequencing platform used, and the criteria used for selecting the oncogenes analyzed. Additionally, it would be useful to provide more information on the criteria used for selecting normal, tumor, and metastatic tissues for analysis.

2.     In section 2.3, the authors describe the protein-protein interaction (PPI) network construction and gene enrichment analysis (GEA) using the STRING and DAVID online tools. The methods used for PPI network construction and GEA are well-described, but it would be helpful to provide more information on the criteria used for selecting enriched PPI clustering networks and the specific statistical methods used for GEA.

3.     In section 2.6, the authors describe the correlation analysis of immune cell infiltration and BIRC5/HIF1Α/FLT4 expressions using the TIMER 2.0 and TISIDB online tools. More information on the criteria used for selecting different molecular subtypes and the specific statistical tests used would be helpful.

4.     In section 2.9, the authors describe the receptor-ligand binding interaction predictions through an in silico molecular docking analysis. The methods used for predicting possible interactions of HN-N07 with its target genes and performing the docking analysis are well-described, but more information on the criteria used for selecting standard inhibitors and the specific statistical tests used would be helpful.

Results

1.     The hierarchical workflow and logic behind decisions at each step are not clearly described. The methodology seems fragmented and lacks a cohesive structure.

2.     The rationale for selection of particular targets and compounds is not well justified. More discussion of target validation and compound design principles is needed.

Discussion and conclusion

Overall, the discussion and conclusion of this study provide a promising avenue for further research into potential small molecule inhibitors of NSCLC progression.

Round 2

Reviewer 1 Report

I regard that the authors have covered me, as they will also publish a second paper with their in vitro data.

-